# Construction of a Tomato (*Solanum lycopersicum* L.) Introgression Line Population and Mapping of Major Agronomic Quantitative Trait Loci

Yifan Chen [1,2], Shuliang Qiu [2], Hui Zhou [2], Wenzheng Gao [2], Lipeng Cui [2], Zhuoyao Qiu [2], Chenchen Dong [2] and Xiaoxuan Wang [2,*]

1   College of Horticulture, Nanjing Agricultural University, Nanjing 210095, China; 2019204030@njau.edu.cn
2   Institute of Vegetables and Flowers, Chinese Academy of Agricultural Sciences, Beijing 100081, China
*   Correspondence: wangxiaoxuan@caas.cn

**Abstract:** Tomato as a fresh fruit has a large market share in China, but few new materials have been developed for such cultivar breeding in recent years. This study aims to create innovative breeding materials for fresh fruit tomatoes with consistent genetic backgrounds and take advantage of beneficial genes from wild germplasm resources. An introgression line (IL) population was constructed using freshly cultivated tomato *S. lycopersicum* 1052 and wild tomato *S. pennellii* LA0716 through hybridization and five consecutive backcrossings, with molecular marker-assisted selection techniques during seedling stages. A total of 447 cleaved amplified polymorphic sequences (CAPS) and 525 simple sequence repeat (SSR) markers were used to screen polymorphic markers among the two parental lines, resulting in 216 polymorphic CAPS and 236 polymorphic SSR markers, with 46.5% parental polymorphism. Then, 200 molecular markers uniformly distributed over the entire genome were further selected, and 107 ILs were finally obtained from 541 BC$_5$ candidate plants. The physical distance between adjacent markers was 6.3~10.0 cm, with an average interval of 7.29 cm, and the IL population constructed covered the whole genome of *S. pennillii* LA0716, with an average introgression segment of 31.5 cm. Moreover, phenotype data of major agronomic traits in BC$_5$ progeny after selfing two times, were analyzed for quantitative trait locus (QTL) mapping, and a total of 11 QTLs distributed on 6 chromosomes were identified, including 3 QTLs regulating plant height, 1 QTL regulating leaf size, 1 QTL regulating fruit color, 4 QTLs regulating fruit weight, and 2 QTLs regulating soluble solids content in ripening fruits. The IL population constructed in this study provided good materials for fresh fruit tomato breeding with improved yield and quality in the future.

**Keywords:** tomato (*Solanum lycopersicum* L.); IL population; agronomic traits; QTL; linkage mapping





## 1. Introduction

Tomato (*Solanum lycopersicum* L.) is the second most important vegetable crop in the world next to potato, which has a high annual yield because of its strong adaptability and easy cultivation. The annual production of fresh tomatoes amounts to approximately 180 million tons in 2019 (www.fao.org/faostat, accessed 1 April 2023). Moreover, tomato is a nutritionally well-balanced food that contains a substantial amount of vitamin A and vitamin C, which plays an important role in global food security and nutrition.

In the process of cultivated tomato breeding, long-term in-species hybridization has been carried out. While screening for high yield and high quality, many beneficial genes against biotic and abiotic stresses have been lost, resulting in lower genetic diversity. A total of 12 closely related species of tomato have been found by now, which all come from the Andean region of South America and belong to the *Solanum* genus in the large *Solanaceae* family [1]. These wild relatives have accumulated many beneficial genes during the evolutionary process, which can provide valuable resources for the genetic improvement of

cultivated tomatoes [2]. However, the method of using the materials obtained from the multi-generational hybridization of cultivated tomato and wild species for genetic improvement does not completely contain all the beneficial genes of wild species [3]. Therefore, an introgression line (IL) population is constructed to contain a single homozygous donor segment in each line and can provide a complete genome in a recurrent background altogether, which not only contributes to the understanding of the genetic control of agronomic traits, but also provides basic materials with a consistent genetic background for the development of new cultivars [4].

The construction of IL populations was originally based on morphological observations, and thus cannot accurately identify the underlying genes of targeted traits. With the development of molecular biology, molecular marker-assisted selection (MAS) was proposed to construct IL populations efficiently, and quantitative trait locus (QTL) mapping techniques have made it easier to identify key alleles involved in agronomic traits [5]. Tomato is one of the first crops used for genes and QTL cloning by a map-based approach [6]. The first IL population in tomatoes originated from a cross between the green-fruited species *Lycopersicon pennellii* (*S. pennellii LA0716*) and the cultivated tomato (cv M82), which consisted of a total of 50 single-segment ILs, with each line containing a single homozygous restriction fragment length polymorphism (RFLP)—a defined *L. pennellii* chromosome segment, providing complete coverage of the wild species genome and a set of lines nearly isogenic to M82 [7]. A field trial of these IL populations and their hybrids revealed at least 23 QTLs for total soluble solids content and 18 for fruit mass, twice as many as previously reported estimates based on traditional mapping of populations [8].

Nowadays, more IL populations of tomato have been constructed for QTL mapping, and near-isogenic lines (NILs) have been established by backcrossing cultivars with QTL that are regarded as simple Mendelian factors, and then map-based gene cloning has been carried out by using the segregating population [9]. Many genes or QTLs detected in tomatoes, e.g., the gene (*Tm-2*) that confers resistance to tobacco mosaic virus, the gene (*wf*) regulating flower color, and the gene (*sun*) regulating fruit shape, were all cloned by using IL populations [3]. However, most of those IL populations were constructed on the basis of tomato cultivar breeding for processed products or small fruits, and few were developed for fresh fruit cultivar breeding, which has a larger market share in China. Moreover, the exchange chromosome segments of the wild-related species used when constructing those IL populations were usually greater than 10 cm, which has some limitations in the further utilization of those IL populations.

Therefore, the objectives of this study were to: (1) screen polymorphic markers from the IL population derived from the cross of *S. lycopersicum* 1052 × *S. pennellii* LA0716 and their $F_1$ progeny; (2) construct genetic linkage maps based on those molecular markers; and (3) identify and map putative QTLs involved in some agronomic traits of tomatoes. The current study innovatively developed the IL lines of *S. pennellii* with the genetic background of fresh fruit tomato, which was rarely reported previously. The results of the current study could be greatly beneficial for improving the efficiency of QTL identification, mapping, cloning, and genetic effect analysis, and further facilitate MAS-based fresh fruit tomato cultivar development in the future.

## 2. Materials and Methods

### 2.1. Plant Materials and DNA Extraction

The IL population was developed through backcrossing five times and selfing two times ($BC_5S_2$) using the fresh fruit tomato inbred line, *S. lycopersicum* 1052, as the recurrent parent, and the green-fruited species *Lycopersicon pennellii*, *S. pennellii* LA0716, as the donor. Specifically, $F_1$ was obtained by crossing 1052 with LA0716 in the Institute of Vegetables and Flowers, Chinese Academy of Agricultural Sciences (CAAS), Beijing, China. *S. Lycopersicum* 1052 is a new excellent fresh fruit tomato breeding line selected by CAAS with characteristics of medium leaf amount, strong growth potential, and high fruit setting rate. The mature fruit of *S. Lycopersicum* 1052 is slightly oblate, pink, about 150 g weight

(single fruit), with medium hardness, without fruit shoulder, and with few abnormal or cracking ones. *S. pennelii* LA1706 originated from Peru and has the characteristics of obtuse and less notched lobules, lower fruit amount under natural conditions, and strong drought tolerance and insect and virus resistance. The fruit of *S. pennelii* LA1706 is smaller than 1 cm in diameter, oblate, and green at maturity (Supplementary Figure S1). *S. pennelii* LA1706 is easy to cross with cultivated parents, but hybridization is one-way. Then, the $BC_1$, $BC_2$, $BC_3$, $BC_4$, and $BC_5$ populations were harvested through backcrossing five times, and then $BC_5S_2$ was obtained by two generations of selfing.

Genomic DNA was extracted from young leaves of 5- to 6-week-old seedlings using the cetyl-trimethylammonium bromide (CTAB) method [10,11] and was diluted to a concentration of 100–150 ng/µL in RNase (10 mg/mL) $H_2O$ (1:100). The integrity and quality of the extracted DNA were evaluated using 2% agarose gel electrophoresis. DNA concentration and OD260/OD280 ratio were determined on an ultraviolet spectrophotometer for quantitative detection of DNA.

### 2.2. Molecular Marker Screening and IL Population Construction

A total of 447 cleaved amplified polymorphic sequence (CAPS) markers and 525 simple sequence repeat (SSR) markers were originally used for genetic analysis of the genomic DNA of parents and $F_1$ plants. The CAPS markers were sourced from Sol Genomics Network (SGN, http://solgenomics.net/, accessed 30 August 2009), and SSR markers were sourced from both SGN (147 markers) and Tomato SBM and Marker Database (http://www.kazusa.or.jp/tomato/, accessed 30 August 2009, 378 markers).

The PCR amplification system for CAPS markers and SSR markers was: 1 µL 100 ng/µL genomic DNA, 5 µL GoTaq® GreenMaster Mix (Promega, Madison, WI, USA), 0.2 µL 10 µmol/L forward primer, 0.2 µL 10 µmol/L reverse primer, and 3.6 µL ddH₂O. The PCR amplification conditions included: initial denaturation at 94 °C for 4 min, followed by 32 cycles of denaturing at 94 °C for 50 s (CAPS markers) or 40 s (SSR markers), and then annealing at 50–60 °C (based on different primers) for 50 s (CAPS markers) or 40 s (SSR markers), extension at 72 °C for 90 s (CAPS markers) or 60 s (SSR markers), single extension at 72 °C for 10 min (CAPS markers) or 7 min (SSR markers), and ended at 16 °C. The CAPS enzyme digestion system was: 10 µL PCR product, 0.2 µL 10 U/µL endonuclease (Takara Biomedical Technology, Beijing, China; NEB, Ipswich, MA, USA), 1.5 µL 10 × Buffer, and 3.6 µL ddH₂O. The enzyme digestion products were detected using 2% agarose gel electrophoresis, and the PCR products with SSR markers were detected using 8% polyacrylamide gel electrophoresis.

Based on the results of polymorphism analysis on the genomic DNA of parents and $F_1$ plants, 200 polymorphic DNA markers that have their genetic locations clearly marked in the genetic linkage map Tomato-EXPEN 2000 (available on the SGN website) [12] and were uniformly distributed on 12 chromosomes of tomatoes were further selected as tracking markers for the IL population construction. These polymorphic DNA markers were then used for whole-genome analysis on 12 $BC_1$ plants, with the primers listed in Supplementary Table S1. The $BC_2$ plants were selected based on the principle of the smallest introgression segment length and number of the donor (*S. pennellii LA0716*) into $BC_1$. For each trace marker, 12 $BC_2$ seedings were sown, constructing a population of 2400 plants of $BC_2$. Starting with the $BC_2$ generation, 200 lines corresponding to the 200 DNA markers with three plants per line were grown and collected for seeds after backcrossing with *S. lycopersicum* 1052. Twelve plants per line per generation were selected for DNA extraction to further screen the corresponding DNA marker until $BC_5$. The proportion of each genotype in each line of the five backcrossing generations was calculated, and deviations from the Mendelian ratio (1:1) were tested using Chi-square analysis [13].

### 2.3. Linkage Map Construction and QTL Mapping

The genetic linkage map used in this study was constructed using the selected DNA markers and the IL population. The positions of the CAPS and SSR markers in linkage



groups and the distance between markers were illustrated through the software GGT2.0 [14] by using the genotype identification results. Briefly, a properly formatted matrix of trait data, including population type, name of individuals, name of markers, map position of markers, and genotypes of individuals (homozygous, hybrid, or missing data) containing column and row headers were copied and pasted from a spreadsheet program into GGT 2.0, and the linkage map overview of the data one chromosome at a time, one genotype at a time, or all chromosomes for all genotypes in a single image was automatically generated in the format of graphical genotypes [15]. Furthermore, a simple correlation analysis of marker data and trait observations was performed by GGT 2.0 for QTL mapping with the interval mapping method using a pair of markers, plotting putative QTL locations along the chromosome bars [14]. The nomenclature of QTL is "q + the abbreviation of trait + the abbreviation of the chromosome where QTL is located + the number of QTL" [16].

### 2.4. Agronomic Trait Investigation and Statistical Analysis

Field investigations were performed to record the phenotype data of the following agronomic traits of plants or fruits in the IL population constructed: plant height (PH, measured from the base of the main stem to the fourth cluster of ripe fruits), leaf size (LS, taken the 5th to 7th real leaf in the same growth period, divided into three categories (large, medium, small) with the medium size leaf from *S. lycopersicum* 1052 as a standard), fruit color (FC, measured by visualization), fruit weight (FW, measured the average weight of all the normal size fruits on the second and third clusters in the ripening period), and soluble solids content in ripe fruits (SSC, measured in triplicates using the digital brix refractometer (PAL-1, Atago, Japan)). The above phenotype data for the agronomic traits were analyzed using SPSS 17.0 (IBM, New York, NY, USA) to check for normality of the data and to test the effects of targeted QTL on different agronomic traits using a *t*-test (with a prior Levene's test for homogeneity of variance) [17]. Moreover, the phenotype of epidermal reticulation (ER) of green fruit was observed in the segregating populations, and the related phenotype data were analyzed using SPSS 17.0 through the Chi-square test.

## 3. Results

### 3.1. Polymorphic DNA Marker Screening

A total of 447 CAPS markers and 525 SSR markers were originally used for genetic analysis on the genomic DNA of parents and $F_1$, and 216 CAPS markers and 236 SSR markers demonstrated polymorphism, accounting for 46.1%. Of those 452 polymorphic markers, 200 markers that were uniformly distributed on the 12 chromosomes were further selected to track chromosome changes during population construction. The summary information, specific locations, and genetic distances of those tracking markers were shown in Table 1 and Supplementary Figure S2. The number of tracking markers located on each chromosome was between 13 and 24, with an average of 16.7. The map distance of adjacent markers was between 6.3 cm to 10.0 cm, with an average of 7.29 cm, and covered 1458 cm of the tomato genome.

Genome-wide genetic analysis of 12 $BC_1$ individual plants using polymorphic markers showed that 49.7% of them were homozygously sourced from *S. Lycopersicum* 1052, 48.1% were heterozygously sourced from both *S. Lycopersicum* 1052 and *S. pennelii* LA0716, and 2.2% were missing. The length of heterozygous chromosomes in 12 $BC_1$ plants was 8408.3 cm, which was 5.8 times as long as the whole genome. After five times backcrossing, the segregation distortion rates of the selected tracking markers in each generation were summarized. All the markers in $BC_1$ followed the Mendelian ratio (1:1). A total of 59 markers showed segregation distortion in $BC_2$ to $BC_5$, with the percentage of segregation distortion for markers in $BC_2$, $BC_3$, $BC_4$, and $BC_5$ being 5.2%, 11.0%, 8.0%, and 12.0%, respectively, among which, five markers exhibited segregation distortion in materials from two generations, and four markers showed segregation distortion in materials from three generations, while no markers displayed segregation distortion in materials from all four generations (Table 2).

**Table 1.** Summary information of the 200 selected tracking molecular markers and their location in the 12 chromosomes of tomatoes during the introgression line population construction.

| Chromosome No. | Number of Polymorphic Markers | Number of Selected Tracking Markers | Chromosome Length (cm) | Average Map Distance of Adjacent Markers (cm) |
|---|---|---|---|---|
| 1 | 40 | 24 | 165 | 6.9 |
| 2 | 44 | 20 | 143 | 7.2 |
| 3 | 78 | 23 | 171 | 7.4 |
| 4 | 58 | 16 | 137 | 8.6 |
| 5 | 33 | 19 | 119 | 6.3 |
| 6 | 22 | 15 | 101 | 6.7 |
| 7 | 25 | 16 | 112 | 7.0 |
| 8 | 21 | 13 | 87 | 6.7 |
| 9 | 33 | 13 | 114 | 8.8 |
| 10 | 27 | 14 | 86 | 6.1 |
| 11 | 22 | 15 | 103 | 6.9 |
| 12 | 49 | 12 | 120 | 10.0 |
| Total | 452 | 200 | 1458 | 7.3 |

**Table 2.** Number of molecular markers that exhibited segregation distortion in plant materials from different backcrossing generations [1].

| Backcrossing Generation [2] | Number of Selected DNA Markers | Number of Markers That Exhibited Segregation Distortion [1] | | Percentage of Segregation Distortion (%) |
|---|---|---|---|---|
| | | $p < 0.05$ | $p < 0.01$ | |
| $BC_1$ | 272 | 0 | 0 | - |
| $BC_2$ | 192 | 9 | 1 | 5.2 |
| $BC_3$ | 182 | 14 | 6 | 11.0 |
| $BC_4$ | 200 | 12 | 6 | 8.0 |
| $BC_5$ | 200 | 18 | 6 | 12.0 |

[1] Tested using Chi-square analysis with *p* value as a criterion. [2] $BC_1$ to $BC_5$: backcrossing generation 1 to 5 using *S. lycopersicum* 1052 as the recurrent parent and *S. pennellii* LA0716 as the donor.

Markers displayed a high skewed frequency of segregation distortion including C2_At4g20410, SSR5, C2_At5g23880, C2_At5g62530, U221657, C2_At4g23840, TG294, C2_At5g19690, and C2_At4g18593. Their locations on chromosomes, segregation distortion generation, and skewed direction are summarized in Table 3.

**Table 3.** The locations on chromosomes, segregation distortion generations, and skewed direction of molecular markers displayed high skewed frequency of segregation distortion.

| DNA Marker | Chromosome | Genetic Distance (cm) | Physical Location on Chromosome | Segregation Distortion Generation [1] | Skewed Direction |
|---|---|---|---|---|---|
| C2_At4g20410 | 2 | 36.9 | q arm | $BC_3$, $BC_4$, $BC_5$ | bidirectional |
| SSR5 | 2 | 53.0 | q arm | $BC_3$, $BC_4$, $BC_5$ | towards the recurrent parent |
| C2_At5g23880 | 3 | 53.5 | around the centromere | $BC_3$, $BC_4$, $BC_5$ | towards the recurrent parent |
| C2_At5g62530 | 6 | 55.5 | q arm | $BC_3$, $BC_5$ | bidirectional |
| U221657 | 8 | 13.0 | around the centromere | $BC_2$, $BC_3$ | bidirectional |
| C2_At4g23840 | 8 | 82.0 | q arm | $BC_4$, $BC_5$ | towards the recurrent parent |
| TG294 | 8 | 87.0 | q arm | $BC_2$, $BC_4$ | towards the recurrent parent |
| C2_At5g19690 | 12 | 27.0 | q arm | $BC_3$, $BC_4$, $BC_5$ | towards the recurrent parent |
| C2_At4g18593 | 12 | 59.0 | p arm | $BC_3$, $BC_4$ | towards the recurrent parent |

[1] $BC_2$ to $BC_5$: backcrossing generation 2 to 5 using *S. lycopersicum* 1052 as the recurrent parent and *S. pennellii* LA0716 as the donor.

### 3.2. IL Population Construction and Linkage Map

Based on the technique of MAS, a total of 541 candidate plants were selected in the population of BC$_5$, which contained 151 single chromosome segments with different lengths that were introgressed from the wild donor, *S. pennellii* LA0716, to the recurrent parent, *S. lycopersicum* 1052. Then, 107 plants were further selected based on the principles of smallest segment length, and overlapped segments that could cover each chromosome. The total length of all the introgression segments was 3368.4 cm, with an average length of 31.5 cm, which totally covered the whole genome of *S. pennillii* LA0716 (1458 cm). A total of 107 ILs were constructed through selfing. The distribution and coverage of the donor introgression segments in the 12 chromosomes were demonstrated in Table 4, and the corresponding linkage map was illustrated in Supplementary Figure S3.

**Table 4.** Summary information of the 200 selected tracking molecular markers and their location on the 12 chromosomes of tomatoes during the introgression line population construction.

| Chromosome No. | Number of Candidate Plants in BC$_5$ Population [1] | Number of Introgression Segments from the Donor | Number of Introgression Segments Selected to Construct the IL Population | Average Map Distance of Adjacent Markers (cm) | | | |
|---|---|---|---|---|---|---|---|
| | | | | Total | Average | Maximum | Minimum |
| 1 | 65 | 17 | 13 | 317.5 | 24.4 | 47.05 | 4.9 |
| 2 | 55 | 15 | 10 | 272.2 | 27.2 | 61.1 | 16.2 |
| 3 | 65 | 21 | 11 | 421.2 | 38.3 | 50.7 | 17.25 |
| 4 | 39 | 12 | 9 | 355.3 | 39.5 | 59.65 | 7.5 |
| 5 | 48 | 12 | 7 | 251.0 | 35.9 | 72.75 | 12.0 |
| 6 | 42 | 10 | 7 | 232.0 | 33.1 | 49.75 | 14.25 |
| 7 | 44 | 16 | 9 | 268.7 | 29.9 | 61.25 | 16.3 |
| 8 | 36 | 10 | 8 | 185.7 | 23.2 | 35.75 | 8.5 |
| 9 | 36 | 9 | 9 | 349.5 | 38.8 | 83.5 | 15.25 |
| 10 | 36 | 9 | 9 | 215.0 | 23.9 | 36.45 | 8.0 |
| 11 | 39 | 10 | 6 | 174.3 | 29.1 | 57.35 | 15.5 |
| 12 | 36 | 10 | 9 | 326.0 | 36.2 | 56.75 | 11.75 |
| Total | 541 | 151 | 107 | 3368.4 | 31.5 | 72.75 | 4.9 |

[1] BC$_5$: backcrossing generation 5 using *S. lycopersicum* 1052 as the recurrent parent and *S. pennellii* LA0716 as the donor.

### 3.3. QTL Identification of the Agronomic Traits

Phenotype data of six agronomic traits were analyzed for QTL mapping, and a total of 12 QTLs distributed on seven chromosomes were identified, which included PH ($n = 3$), LS ($n = 1$), FC ($n = 1$), FW ($n = 4$), SSC ($n = 2$), and ER ($n = 1$) at the $p < 0.05$ level. Specifically, the three PH QTLs are located on chromosomes 2, 3, and 7, and were named *qPH2a*, *qPH3a*, and *qPH7a*, with *qPH2a* and *qPH3a* decreasing the PH by 15% to 27% and 17% to 25%, respectively, while *qPH7a* increased the PH by 12% to 24%. The LS QTL and FC QTL are located on chromosomes 12 and 3, are named *qLS12a* and *qFC3a*, and correspond to larger LS and yellow fruit, respectively. The four FW QTLs are located on chromosomes 1, 2, and 3, are named *qFW1a*, *qFW2a*, *qFW3a*, and *qFW3b*, and decreased FW by 18% to 30%, 23% to 50%, 12% to 28%, and 24% to 27%, respectively. The two SSC QTLs are located on chromosomes 7 and 9, are named *qSSC7a* and *qSSC9a*, and increased SSC by 16% to 33% and 22% to 30%, respectively. The QTL controlling ER is located on chromosome 4 and is named *qER4a*. The specific physical positions and the mapped markers upstream and downstream are shown in Table 5.

**Table 5.** Putative quantitative trait loci (QTLs) detected for the major agronomic traits of tomatoes in the constructed introgression line (IL) population.

| Agronomic Trait [1] | QTL | Chromosome | Physical Position | Marker Upstream | Marker Downstream | Molecular/ Physiological Role | *p*-Value [2] |
|---|---|---|---|---|---|---|---|
| PH | *qPH2a* | 2 | SL2.50ch02:0..37,699,910 | C2_At5g37260 | SSR40 | decreased PH by 15–27% | 0.042 |
| | *qPH3a* | 3 | SL2.50ch03:61,948,912..65933054 | C2_At1g05330 | C2_At1g52590 | decreased PH by 17–25% | <0.001 |
| | *qPH7a* | 7 | SL2.50ch07:65,185,067..67,592,440 | C2_At3g14910 | C2_At5g56130 | increase PH by 12–24% | 0.001 |
| LS | *qLS12a* | 12 | SL2.50ch12:5,277,091..63,005,148 | C2_At5g42740 | T0801 | larger LS | <0.001 |
| FC | *qFC3a* | 3 | SL2.50ch03:3,464,378..3,465,245 | T1388 | cLPT-2-E21 | yellow fruit | <0.001 |
| FW | *qFW1a* | 1 | SL2.50ch01:79,707,633..87,866,313 | C2_At4g15520 | U223116 | decrease FW by 18–30% | 0.01 |
| | *qFW2a* | 2 | SL2.50ch02:50,645,729..52,761,764 | T1480 | U153274 | decrease FW by 23–50% | 0.01 |
| | *qFW3a* | 3 | SL2.50ch03:1,755,716..16,135,852 | TG130 | T1388 | decrease FW by 12–28% | 0.016 |
| | *qFW3b* | 3 | SL2.50ch03:59,592,414..67,546,853 | C2_At3g12490 | C2_At3g17970 | decrease FW by 24–27% | 0.023 |
| SSC | *qSSC7a* | 7 | SL2.50ch07:66,922,790 | C2_At4g26750 | - | increase SSC by 16–33% | 0.008 |
| | *qSSC9a* | 9 | SL2.50ch09:1,916,815..3,791,130 | C2_At2g32600 | C2_At2g37500 | increase SSC by 22–30% | <0.001 |
| ER | *qER4a* | 4 | SL2.50ch04:62,469,833.. 65,801,303 | SSR214 | C2_At1g76080 | corky and reticulated epidermis | <0.05 |

[1] FC: fruit color; FW: fruit weight; LS: leaf size; PH: plant height; SSC: soluble solids content; ER: epidermal reticulation. [2] Results from the *t*-test by comparing phenotype data from plants with or without such QTL.

## 4. Discussion

The first objective of this study was to construct the IL population to screen polymorphic markers. The IL population was different only in a few chromosomal regions, which could effectively reduce the interference of genetic background variation, eliminate the epistasis effect of donor parents and improve the identification probability of minor effect QTL [8]. The IL population, constructed by Eshed and Zamir [7], which used processed tomato species cv M82 and *S. pennellii* as the genetic background, is the most widely used IL population in tomatoes by far, for the detection of QTL and discovery of candidate genes. This population initially consisted of 50 lines, and later increased to 76 lines. In the current study, the excellent freshly cultivated tomato, *S. lycopersicum* 1052, and the wild tomato, *S. pennellii* LA0716, were used to construct the pennellii IL population through hybridization, five consecutive backcrossings, five molecular marker-assisted selections during seedling stages, and two generations of selfing. The whole-genome sequencing on $BC_1$ showed that the length of the heterozygous chromosomes in $BC_1$ was about six times as long as the whole genome; moreover, the percentage of segregation distortion for 200 markers in $BC_1$ to $BC_5$ was 0~12.0%, greatly lower than the measured percentage of 68% reported by Robertson et al. [18], indicating that the recurrent parent genome and genes were sufficiently recovered. A total of nine locations were discovered as the hot spots for segregation distortion, which may be related to the male sterile (*ms*) genes, for example, C2_At4g20410 and SSR5 may be related to *ms-2*, *ms-5*, *ms-10*, *ms-15*, and *ms-26* located on chromosome 2; C2_At5g23880 may be related to *ms-9* located on chromosome 3; C2_At5g62530 may be related to *ms-16* and *ms-33* located on chromosome 6; U221657, C2_At4g23840 and TG294 may be related to *ms-8* and *ms-17* located on chromosome 8. Compared with the existing IL populations constructed for processed tomato or small fruit tomato breeding, e.g., *S. pennellii* LA0716, *S. habrochaites* LA1777, *S. lycopersicoides* LA2951, and *S. habrochaites* LYC4, this population built in the current study contained more lines and had more and smaller fragments on each chromosome on average, which is conducive to more accurate positioning of genetic loci. Moreover, these lines containing specific introgressed fragments could be directly used to shorten the breeding period during fresh fruit cultivars' development, providing good materials for fresh tomato breeding.

The second objective of this study was to construct genetic linkage maps based on the selected molecular markers. In the current study, CAPS and SSR molecular markers were

used to screen targeted plants in each generation during IL population construction. The CAPS markers use specific primers to amplify specific DNA fragments and then use specific restriction enzymes to generate polymorphisms. The SSR markers can amplify DNA fragments of different lengths through the complementary sequences at both ends of the repeat sequence. Both CAPS and SSR are co-dominant markers with low requirements for DNA quality and can be detected by PCR amplification and agarose gel or polyacrylamide gel electrophoresis, which is economical and convenient. In comparison, previous IL populations were mainly constructed using RFLP or amplified fragment length polymorphism (AFLP) markers, which had high requirements for DNA quality, and were complicated, costly, time-consuming, and required fluorescent or isotopic labeling, making it difficult to select a large number of individual plants. The 200 molecular markers selected for tracking in the current study were all located on the Tomato-EXPEN 2000 genetic linkage map constructed by *S. lycopersicum* LA925 × *S. pennellii* LA0716 and had specific genetic positions, which has laid a good foundation for the application of this population in the future.

The third objective of this study was to identify and map putative QTLs involved in some agronomic traits. Based on the observation and statistical analysis of the phenotype data in the IL population constructed in the current study, it was discovered that the population contained many excellent agronomic traits. For example, regarding PH, a line with significantly reduced internode length was found in the population; regarding fruit color, the population contained red, orange-red, pink and yellow fruits; regarding SSC, most fruits in this population had SSC distributed between 3.0 and 7.0, some even reached 8.0. Because there were fewer exogenous introgressed fragments in the IL population constructed, the excellent plant materials selected from the population could be directly used to facilitate the actual breeding work.

In the current study, the IL population constructed by *S. pennellii* was used for the first time to discover the QTLs that are highly related to the PH in tomatoes. The three loci detected have not been reported in previous studies, two of which showed a negative regulatory effect on plant height of the 4th cluster of the fruit, indicating important implications in plant growth regulation of tomato. There were few previous studies that reported QTL screening on the height of tomato plants, especially for the height of the 4th cluster of fruit, which is of great significance in actual tomato production. Grandillo and Tanksley [19] selected plants from the $BC_2S_1$ and $BC_3$ of the IL population sourced from *L. pirnpinellifolium* (LA1589) × M82, and detected multiple loci that were highly involved in PH on chromosomes 1, 5, 7, 8, 9, and 11. Moreover, Prudent et al. [20] detected multiple loci on chromosomes 3, 4, 9, 11, and 12 that were highly associated with the PH of the 4th cluster of fruit, using an IL population constructed with *S. chmielewskii* LA1840 and Moneyberg.

Leaf size is also an important agronomic trait that would affect plant growth through photosynthesis regulation. In this study, a QTL locus *qLS12a* was detected using the IL population built from the *S. pennellii* IL population on chromosome 12, which showed a positive regulatory effect on LS. Holtan and Hake [21] detected eight QTLs that could negatively affect LS using an IL population constructed by *S. pennellii*. In addition, Prudent et al. [20] found three QTLs that were associated with the total leaf area using an IL population constructed by *S. chmielewskii*, of which two loci increased the total leaf area.

The fruit colors of the parents in the constructed IL population, 1052 and LA0716, were pink and green, respectively. The physical location of the detected QTL *qFC3a* (SL2.50ch03:3464378..3465245) that related to yellow fruit was very close to that of the *r* gene (SL2.50ch03:4325332..4330923) (about 0.73 Mb), which was reported to inhibit the transcription of PSY1 gene and make the fruit yellow [22]. The fact that the *qFC3a* and *r* genes did not exactly match may be caused by the low density of markers on chromosome 3 and the introgression of the exogenous fragments not precise enough.

The QTLs controlling FW have been widely reported. So far, 117 QTLs related to FW have been unearthed from IL populations constructed from seven wild tomato cultivars,

including *S. chmielewskii*, *S. cheesmaniii*, *S. hirsutum*, *S. parviflorum*, *S. peruvianum*, *S. pimpinellifolium* and *S. pennellii*, among which at least 28 QTLs were detected in two or more IL populations, and 23 QTLs were detected in *S. pennellii* IL populations [23,24], with two loci *fw2.2* and *fw3.2* having been cloned [9,25]. In this study, four QTLs related to FW were found using the *S. pennellii* IL population with more lines and shorter exogenous fragments, among which *qFW3a* reduced fruit weight by an average of 17%, which was a new QTL beyond the four loci related to FW on chromosome 3 reported previously according to its physical position. Gandillo et al. [23] detected the fruit size-related locus *fw1.2* using the BC₁ population constructed by *S. pimpinellifolium* and *S. lycopersicum*, and anchored it within a physical distance of 8.6 Mb. Furthermore, Lippman and Tanksley [26] anchored *fw1.2* within the range of 14.5 Mb using the F₂ population constructed by *S. pimpinellifolium* and *S. lycopersicu*. The physical position of *qFW1a*, a locus related to FW found in this study, coincided with the position of *fw1.2*, and further narrowed the positioning range of this QTL to 7.7 Mb. Two other major QTLs detected in this study regulating FW, *qFW2a* and *qFW3b*, were also consistent with the two cloned genes reported previously. *qFW2a* reduced FW by 23% to 50% and was the same as the cloned gene *fw2.2*, which was reported to regulate the size of plants and organs by regulating the number of cells [9,25]. *qFW3b* reduced FW by 24% to 27%, and its physical location was very close to the cloned gene *fw3.2*, only 0.7 Mb away, which was reported to control the fruit size by regulating the number of cells in the pericarp and placental tissue and affecting the time of fruit ripening [25]. The reason why *qFW3a* and fw3.2 did not fully match may be due to the limited labeling density in this study.

There were also some reports on QTLs controlling SSC in fruits previously. So far, 95 QTLs associated with SSC in tomato fruits have been discovered using 14 IL populations constructed from eight wild tomato cultivars, including *S. cheesmaniae*, *S. chmielewskii*, *S. habrochaites*, *S. neorickii*, *S. pimpinellifolium*, *S. pennelli*, *S. arcanum* and *S. lycopersicum*, among which 7 and 11 loci were located on chromosomes 7 and 9, respectively [27]. In most cases, the SSC in fruits is negatively correlated with the fruit size [23]. In this study, two QTLs, *qSSC7a* and *qSSC9a*, related to SSC, were detected using the *S. pennellii* IL population with more lines and shorter exogenous fragments. Both loci significantly increased the SSC of the fruit under the condition of a low reduction in FW, which is of great value for improving fruit quality. As a newly discovered locus by comparing the physical location of *qSSC7a* with the QTLs on chromosome 7 reported before, *qSSC7a* increased the SSC by 16% to 32% and reduced FW by 6% to 13%. Thus, lines containing *qSSC7a* have uniform fruit size, and could be treated as excellent breeding materials for high-quality tomato fruits. *qSSC9a* matched the position of the previously cloned gene *brix9-2-5*, which influenced sucrose transport by changing the activity of sucrose invertase in the apoplast, thereby regulating the SSC in fruits [28,29].

Fruit cracking is a physiological disorder associated with fractures in the fruit cuticle, and the studies on QTLs regulating fruit-cracking started in the 1950s, but the progress was slow, especially for tomatoes, in which the genetics of fruit cracking resistance are extremely complex, e.g., different types of tomato fruit cracking were reported to be controlled by different QTLs [30]. In the current study, a tomato fruit cracking type referred to as epidermal reticulation (which belongs to cuticle cracking) was observed, the fruits which initially appeared fissure at 23 days after pollination, and then were characterized by the corky, reticulated epidermis in fruits at the mature green stage, lastly resembled the "melon-like" skin on the mature fruit. The QTL regulating ER is located on chromosome 4, which was also reported previously [31,32].

The IL populations construction and putative QTL discovery in the current study were only an initial step for breeding new cultivars. In the next step, more work will be conducted, including exploration of the gene/allele underlying each putative QTL through fine mapping and map-based gene cloning, manipulation of candidate genes through targeted gene editing of *cis*-regulatory or coding sequences using techniques such

as CRISPR/Cas9, and validation of targeted gene function through field trials [4], with the ultimate goal of creating new cultivars and fully exploiting those IL populations.

**5. Conclusions**

In conclusion, a population of 107 ILs was finally constructed by fresh cultivated tomato *S. lycopersicum* 1052 × wild tomato *S. pennellii* LA0716 after five consecutive back-crossings and molecular marker-assisted selection using 200 CAPS and SSR markers. In this population, the average physical distance between adjacent markers was 7.29 cm, and the average introgression segment was 31.5 cm, which could cover the whole genome of *S. pennillii* LA0716. A total of 12 QTLs distributed on seven chromosomes were mapped, which could regulate major agronomic traits including plant height, leaf size, fruit color, fruit weight, soluble solids content in ripen fruits, and epidermal reticulation in green fruit. The IL population constructed in this study provided good materials for fresh tomato breeding, but more related work on QTL fine mapping is still needed in the future.

**Supplementary Materials:** The following supporting information can be downloaded at: https://www.mdpi.com/article/10.3390/horticulturae9070823/s1, Table S1: Primer sequences of the markers used to confirm polymorphism in the current study. Figure S1: Field phenotypes of *S. lycopersicum* 1052 (left) and *S. pennellii* LA0716 (right) for introgression lines construction in the current study. Figure S2: Specific locations and genetic distances of the 200 selected tracking molecular markers in the 12 chromosomes of tomatoes during the introgression line population construction; Figure S3: Linkage map of selected chromosome segments introgressed from the wild donor (segments labelled by grey shadow), *S. pennellii* LA0716, to the recurrent parent (segments labelled with slash lines), *S. lycopersicum* 1052 for introgression lines construction.

**Author Contributions:** Conceptualization, X.W.; methodology, X.W., Y.C., S.Q., H.Z. and L.C.; investigation, Y.C., S.Q., H.Z., W.G., L.C., Z.Q. and C.D.; data curation, Y.C., S.Q., H.Z., W.G. and L.C.; writing—original draft preparation, Y.C.; writing—review and editing, X.W.; supervision, X.W.; funding acquisition, X.W. and W.G. All authors have read and agreed to the published version of the manuscript.

**Funding:** This research was funded by the Hainan Special PhD Scientific Research Foundation of Sanya Yazhou Bay Science and Technology City (Grant number HSPHDSRF-2022-10-005), the earmarked fund for China Agriculture Research System (CARS) (Grant number CARS-23-A06), and the National Natural Science Foundation of China (Grant number 31672153).

**Data Availability Statement:** The data presented in this study are available on request from the corresponding author. The data are not publicly available due to the requirements of the current project.

**Acknowledgments:** Authors acknowledge research participants, including laboratory technicians for contributing their time and experiences for this study.

**Conflicts of Interest:** The authors declare no conflict of interest.

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
