# Peer review of "Construction of a Tomato (Solanum lycopersicum L.) Introgression Line Population and Mapping of Major Agronomic Quantitative Trait Loci"

_horticulturae, doi:10.3390/horticulturae9070823_

Round 1

Reviewer 1 Report

The study screened polymorphic markers of population derived from the cross of S. lycopersicum species to construct genetic linkage maps based on those molecular markers identifing some putative QTLs involved in some agronomical traits of tomatoes.  The main finds of study is the  the indentification of  agronomical traits that could improve the molecular marker-assisted selection of tomato cultivars.

Overall the study is well structured and organized. There is a few of scope of improvement in this paper. The major critic point would be the tables that can be improved. The discussion of the data is well supported, although the molecular and physiological role of the QTLs identified on tomato breeding could be better highlighted.  

Overall the language is good, but for some points minor editing of English language is required.

Author Response

Response: Thanks very much for your valuable comments. We have made some changes to the tables especially the footnotes to make the tables more understandable. Moreover, we have changed the structures of the discussion related to the identified QTLs to make their molecular and physiological roles better highlighted. Lastly, we have had the professional agent to help revising the English of the whole manuscript. All the changes have been marked up using the “Track Changes” function, which we hope can be approved.

Reviewer 2 Report

I read the manuscript Horticulturae-2428767 with interest.

It brings interesting data to the S. lycopersicum QTL mapping and molecular markers assisted selection.

The introduction introduces the planned scope of the research and indicates the objective. 

I like that the objectives are summarized in points at the end of the introduction,

however, I miss the systematic explanation of these objectives in the chapter of discussion

The material and methods are correctly presented with the exception of detailed description of bioinformatic methods.

I suggest to summarize briefly the operating principle of the software and workflow used with GGT2.0

The results were presented correctly and I have no comments.

Discussion carried out correctly

Conclusions refer to the research carried out

References selected and used correctly

Specific comments:

Please clearly describe what is the new in this study and how the results can be used from a practical point of view. I suggest to explain more in the practical usefulness of this kind of tests and supporting them with the scientific literature  (e.g. in the introduction section)

In addition to all this, I believe that this study is an initial step of the project. And since the QTL study is the early study of genes couse the trait variations, in order to identify the actual genes, I would consider it necessary to briefly detail the methodology by which you plan to study the gene identification in the following and how the here presented results can be used in the planned gene identification processes.

Author Response

Response to Comments of Reviewer 2

I read the manuscript Horticulturae-2428767 with interest. It brings interesting data to the S. lycopersicum QTL mapping and molecular markers assisted selection. The introduction introduces the planned scope of the research and indicates the objective. I like that the objectives are summarized in points at the end of the introduction, however, I miss the systematic explanation of these objectives in the chapter of discussion.

Response: Thanks very much for your comments. We have restated the corresponding objectives in the beginning of each paragraph of the discussion chapter to enhance the systematicity of this section.

The material and methods are correctly presented with the exception of detailed description of bioinformatic methods. I suggest to summarize briefly the operating principle of the software and workflow used with GGT2.0.

Response: Thanks very much for your comments. The operating principles and workflow of GGT2.0 have been summarized in details as shown in L156-165.

The results were presented correctly and I have no comments. Discussion carried out correctly. Conclusions refer to the research carried out. References selected and used correctly.

Response: Thanks very much for your approval of those sections.

Specific comments:

Please clearly describe what is the new in this study and how the results can be used from a practical point of view. I suggest to explain more in the practical usefulness of this kind of tests and supporting them with the scientific literature (e.g. in the introduction section)

Response: Thanks very much for your comments. We have re-emphasized the novelty of this study and the practical application of the results with scientific literatures supporting in the introduction section. Please see L73-78 and L88-91.

In addition to all this, I believe that this study is an initial step of the project. And since the QTL study is the early study of genes cause the trait variations, in order to identify the actual genes, I would consider it necessary to briefly detail the methodology by which you plan to study the gene identification in the following and how the here presented results can be used in the planned gene identification processes.

Response: Thanks very much for your comments. We have added the detailed information including methodologies we plan to do in the next step for genes identification and the utilization of the present results in this process. Please see L402-409.

Reviewer 3 Report

I presume the Authors put a lot of work into carrying out this important experiment, which works to the advantage of this manuscript. Unfortunately, the description of the experiment is insufficient and incomplete.

Authors did not describe the F1 development, how was the parents evaluated. What are the characteristics of the parents used and what did they contribute to the F1 hybrid. These things are lacking.

Backcrossing of BC1 – BC5 was not discussed in the manuscript. What about the recurrent parent genome recovery.

The QTLs identified, authors did not report whether the recurrent parent genes were sufficiently recovered or not.

Authors did not compare their results to the standard Mendelian laws. This is not acceptable especially in plant breeding and genetics. Does the current result agree with mendel or not?

No information on the primer sequence of the markers used to confirm polymorphism. Authors should report the primer sequence of the 200 polymorphic markers and that could be part of the supplementary materials.

The photos and graphics quality presented were extremely poor and makes it impossible to read especially in figure S2 supplementary materials.

Still poor description of the genotypic selection in both M&M and results.

Minor

Abstract

To explore and utilize the beneficial genes of wild germplasm resources, innovate breed- 10 ing materials with consistent genetic background, and develop new cultivars with improved yield 11 and quality of tomato, an introgression line (IL) population was constructed using fresh cultivated 12 tomato S. lycopersicum 1052 and wild tomato S. pennellii LA0716 through 1 crossing, 5 consecutive 13 backcrossing and molecular marker-assisted selection during seedling stages………………recast this statement. Use short sentences to avoid errors.

Moderate editing of English language required.

Author Response

Response to Comments of Reviewer 3

I presume the Authors put a lot of work into carrying out this important experiment, which works to the advantage of this manuscript. Unfortunately, the description of the experiment is insufficient and incomplete.

Response: Thanks very much for your comments. We have added more information according to your comments to make the description of the experiment more sufficient and more complete. Please see the following responses.

Authors did not describe the F1 development, how was the parents evaluated. What are the characteristics of the parents used and what did they contribute to the F1 hybrid. These things are lacking.

Response: Thanks very much for your comments. The characteristics of the parents and the parents’ evaluation during F1 development have been described in details in the M&M section. Please see L100-109.

Backcrossing of BC1 – BC5 was not discussed in the manuscript. What about the recurrent parent genome recovery.

Response: Thanks very much for your comments. We have added the discussion about backcrossing of BC1 – BC5. Moreover, the whole-genome sequencing analysis results of BC1 have been added, together with the percentage of segregation distortion for 200 markers in BC1 to BC5, all indicating the recurrent parent genome were sufficiently recovered. The above information has been added in the results and discussion sections. Please see L198-202, and L272-283.

The QTLs identified, authors did not report whether the recurrent parent genes were sufficiently recovered or not.

Response: Thanks very much for your comments. The whole-genome sequencing analysis results of BC1 have been added, together with the percentage of segregation distortion for 200 markers in BC1 to BC5, all indicating the recurrent parent genes were sufficiently recovered. The above information has been added in the results section and has been discussed. Please see L198-202, and L272-283.

Authors did not compare their results to the standard Mendelian laws. This is not acceptable especially in plant breeding and genetics. Does the current result agree with mendel or not?

Response: Thanks very much for your comments. The comparison between our results and the standard Mendelian laws was showed in details in the results section and in Table 2. We have added some discussion on these results. Please see L272-283.

No information on the primer sequence of the markers used to confirm polymorphism. Authors should report the primer sequence of the 200 polymorphic markers and that could be part of the supplementary materials.

Response: Thanks very much for your comments. We have added the primer sequences of the markers used to confirm polymorphism in the current study as Supplementary Table 1.

The photos and graphics quality presented were extremely poor and makes it impossible to read especially in figure S2 supplementary materials.

Response: Thanks very much for your comments. We have re-drawn all the supplementary figures to make them clearer and more readable.

Still poor description of the genotypic selection in both M&M and results.

Response: Thanks very much for your comments. We have added more information according to your comments to make the description of the genotypic selection processes more sufficient and more complete. Please see the above responses.

Minor

Abstract

To explore and utilize the beneficial genes of wild germplasm resources, innovate breeding materials with consistent genetic background, and develop new cultivars with improved yield and quality of tomato, an introgression line (IL) population was constructed using fresh cultivated tomato S. lycopersicum 1052 and wild tomato S. pennellii LA0716 through 1 crossing, 5 consecutive backcrossing and molecular marker-assisted selection during seedling stages………………recast this statement. Use short sentences to avoid errors.

Response: Thanks very much for your comments. We have revised this statement with shorter sentences. Please see the abstract section (L10-15).

Comments on the Quality of English Language

Moderate editing of English language required.

Response: Thanks very much for your comments. We have had the professional agent to help revising the English of the whole manuscript. All the changes have been marked up using the “Track Changes” function, which we hope can be approved.

Round 2

Reviewer 3 Report

Minor corrections

Abstract 

through 1 crossing and 5 consecutive backcrossings………..through hybridization and five consecutive backcrossings

2 times selfing……….two times selfing

Also, minor English check is recommended.

Author Response

Dear editors and reviewer:

Thank you again for your letter and the reviewer 3’s comments concerning our manuscript entitled “Construction of tomato (Solanum lycopersicum L.) introgression line population and mapping of major agronomic quantitative trait loci”. (Manuscript ID horticulturae-2428767). These comments are very valuable and helpful for further revising and improving our manuscript. We have made corrections to the manuscript accordingly. Moreover, we have had a professional agent to further help revise the English of the whole manuscript. All the changes have been marked up using the “Track Changes” function, which we hope can be approved. Thanks again for your effort in processing our manuscript.

Sincerely,

Yifan Chen

Response to Comments from Reviewer 3

Minor corrections

Abstract

through 1 crossing and 5 consecutive backcrossings………..through hybridization and five consecutive backcrossings

2 times selfing……….two times selfing

Response: Thank you very much for your suggestions. We have revised these statements accordingly. Please see the abstract section and the other sections in the manuscript.

Comments on the Quality of English Language

Also, minor English check is recommended.

Response: Thank you very much for your suggestion. We have had a professional agent to further help revise the English of the whole manuscript. All the changes have been marked up using the “Track Changes” function, which we hope can be approved.